# Continual Panoptic Perception: Towards Multi-modal Incremental Interpretation of Remote Sensing Images

Bo Yuan
Beihang University
Beijing, China
Tianmushan Laboratory
Hangzhou, China
yuanbobuaa@buaa.edu.cn

Danpei Zhao*
Beihang University
Beijing, China
Tianmushan Laboratory
Hangzhou, China
zhaodanpei@buaa.edu.cn

Zhuoran Liu
Beihang University
Beijing, China
liuzhuoran@buaa.edu.cn

Wentao Li
Beihang University
Beijing, China
canoe@buaa.edu.cn

Tian Li
Beihang University
Beijing, China
Tianmushan Laboratory
Hangzhou, China
lit@buaa.edu.cn

## Abstract

Continual learning (CL) breaks off the one-way training manner and enables a model to adapt to new data, semantics and tasks continuously. However, current CL methods mainly focus on single tasks. Besides, CL models are plagued by catastrophic forgetting and semantic drift since the lack of old data, which often occurs in remote-sensing interpretation due to the intricate fine-grained semantics. In this paper, we propose Continual Panoptic Perception (CPP), a unified continual learning model that leverages multi-task joint learning covering pixel-level classification, instance-level segmentation and image-level perception for universal interpretation in remote sensing images. Concretely, we propose a collaborative cross-modal encoder (CCE) to extract the input image features, which supports pixel classification and caption generation synchronously. To inherit the knowledge from the old model without exemplar memory, we propose a task-interactive knowledge distillation (TKD) method, which leverages cross-modal optimization and task-asymmetric pseudo-labeling (TPL) to alleviate catastrophic forgetting. Furthermore, we also propose a joint optimization mechanism to achieve end-to-end multi-modal panoptic perception. Experimental results on the fine-grained panoptic perception dataset validate the effectiveness of the proposed model, and also prove that joint optimization can boost sub-task CL efficiency with over 13% relative improvement on panoptic quality. The project page is available at https://github.com/YBIO/CPP.

*Corresponding author

## CCS Concepts

• **Computing methodologies** → **Computer vision tasks**.

## Keywords

Continual Learning, Panoptic Perception, Multi-task Learning, Remote-sensing Interpretation, Catastrophic Forgetting

**ACM Reference Format:**
Bo Yuan, Danpei Zhao, Zhuoran Liu, Wentao Li, and Tian Li. 2024. Continual Panoptic Perception: Towards Multi-modal Incremental Interpretation of Remote Sensing Images. In *Proceedings of the 32nd ACM International Conference on Multimedia (MM '24), October 28–November 1, 2024, Melbourne, VIC, Australia.* ACM, New York, NY, USA, 10 pages. https://doi.org/10.1145/3664647.3680654

## 1 Introduction

Continual learning (CL) enables a model to continuously acquire knowledge in a sequential manner. Currently, many CL methods are designed for specific single tasks, which cover natural language processing [27], computer vision [28, 58, 65], etc. However, the complex and practical applications urge the model to have the capacity for CL in multi-task learning (MTL). For example, the continuously incremental data in remote sensing usually requires the model to have the ability to continually interpretation on new data, semantics and tasks.

Over the past decade, CL has been intensively concerned since it can break through the typical one-off training schema and enable the model to evolve with continuous data. However, CL encounters two main challenges including catastrophic forgetting and semantic drift [56, 65]. These problems occur when the parameter updates result in the loss of previously learned knowledge, leading to prediction chaos and model degradation. Traditionally, the popular fully-supervised methods conduct a complete re-training on the incremental data that may result in an analogous issue akin to Alzheimer's disease, where the model tends to forget its previously learned knowledge due to parameter changes [28]. However, current CL approaches face challenges in the trade-off between

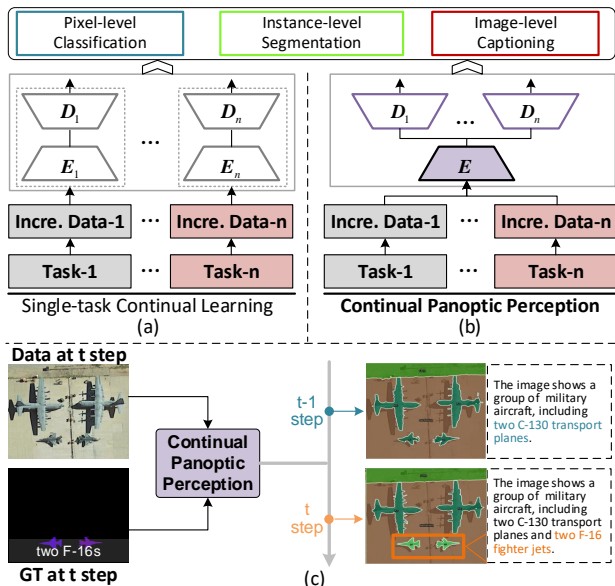

**Figure 1: The proposed Continual Panoptic Perception (CPP) architecture. (a): Single-task CL methods only support separate training on different tasks. (b): CPP enables a shared encoder across multi-modal tasks, which also supports multi-task continual learning within a single model. (c): CPP achieves class-incremental pixel classification, instance segmentation and image captioning.**

preserving old knowledge and learning new ones. A range of researches [3, 4, 9] propose to retrospect known knowledge including sample selection as exemplar memory [43, 50]. However, these replay-based methods usually bring extra memory costs and arouse privacy concerns. Another kind that does not rely on old data utilizes transfer learning manners like knowledge distillation to inherit the capability of the old model [46, 53]. In remote-sensing images (RSIs), the complex semantic relations and fine-grained semantic classes make CL extremely challenging. And there is still a lack of exploration on multi-task and multi-modal CL.

As seen in Fig. 1(a), typical single-task CL approaches only support separate training on different tasks, which limits the CL capacity on complex practical scenarios. In this paper, we propose a CL architecture for panoptic perception [64] namely Continual Panoramic Perception (CPP). As illustrated in Fig. 1(b), CPP enables multi-task CL within a single model that shares the same image encoder for multi-modal interpretation. Particularly, CPP consists of a collaborative cross-modal encoder (CCE), a task-interactive knowledge distillation (TKD) module and a task-asymmetric pseudo-labeling (TPL) mechanism. Concretely, the CCE module extracts image features, which are projected to mask embeddings and text embeddings synchronously for the corresponding decoder branch. To alleviate catastrophic forgetting, the proposed TKD module utilizes multi-modal features to address semantic drift to boost distillation efficiency. While TPL integrates the pseudo label to improve the label confidence for CL steps. Fig. 1(c) illustrates the proposed CPP training pattern that synchronously proceeds segmentation

and captioning tasks within an integrated model. After each CL step, new semantics are learned while achieving compatibility with the old knowledge.

The main contributions of this paper are summarized as follows:

- We propose continual panoptic perception, a novel architecture supporting multi-task continual learning covering pixel-level classification, instance-level segmentation and image-level captioning.
- We present a task-interactive knowledge distillation method, utilizing cross-task knowledge inheritance and task-asymmetric pseudo-labeling to reconcile stability and plasticity.
- The proposed method achieves an end-to-end continual learning manner on remote-sensing panoptic perception dataset, and also proves the feasibility of joint optimization across multi-modal CL tasks.

## 2 Related Work

### 2.1 Continual Image Segmentation

Continual learning originates from as early as [34] and has been explored in various fields, including computer vision [65], natural language processing [60], remote-sensing [33], etc. With respect to image segmentation, the main challenges are catastrophic forgetting and semantic drift, which arise from the absence of old data and parameter updates [8, 22]. According to whether relying on old data [56], the CL methods can be divided into replay-based methods [7, 25, 29, 50, 58, 63] and exemplar-free methods [2, 6, 17, 35, 39, 44, 46, 53, 59, 65]. The former involves storing a portion of past training data or features as exemplar memory, which brings extra memory costs and raises privacy concerns. The latter usually utilizes knowledge distillation to inherit the capability of the old model. Considering reducing reliance on annotations, few-shot CL methods are also deeply explored in [24, 40, 47, 66]. However, current CL methods are mainly designed for single tasks, the multi-task continual learning has not been deeply studied.

### 2.2 Image Captioning

Image captioning [20] describes an image with words or sentences to meet human-like semantic understanding. Depending on the network architecture, there are main three branches including heterogeneous-architecture-based, attention-based, and pre-training-based methods. Encoder-decoder architectures normally include a CNN-based visual encoder, an RNN-based language decoder, and a cross-modal attention block [19, 54]. Attention-based methods [1, 23] introduce semantic attention mechanisms to model contextual information among areas in an image. While some recent methods witness large performance boosting via large-scale pre-training [10, 21] and promoting [52]. Continual image captioning allows generating captions over a series of new tasks coming with new semantics [15, 37]. Considering the rich semantics in RSIs, the investigation of continual image caption is a valuable and promising task.

### 2.3 Multi-task Learning

Multi-task learning (MTL) aims to jointly learn multiple related prediction tasks with shared information across tasks [45, 61, 62].

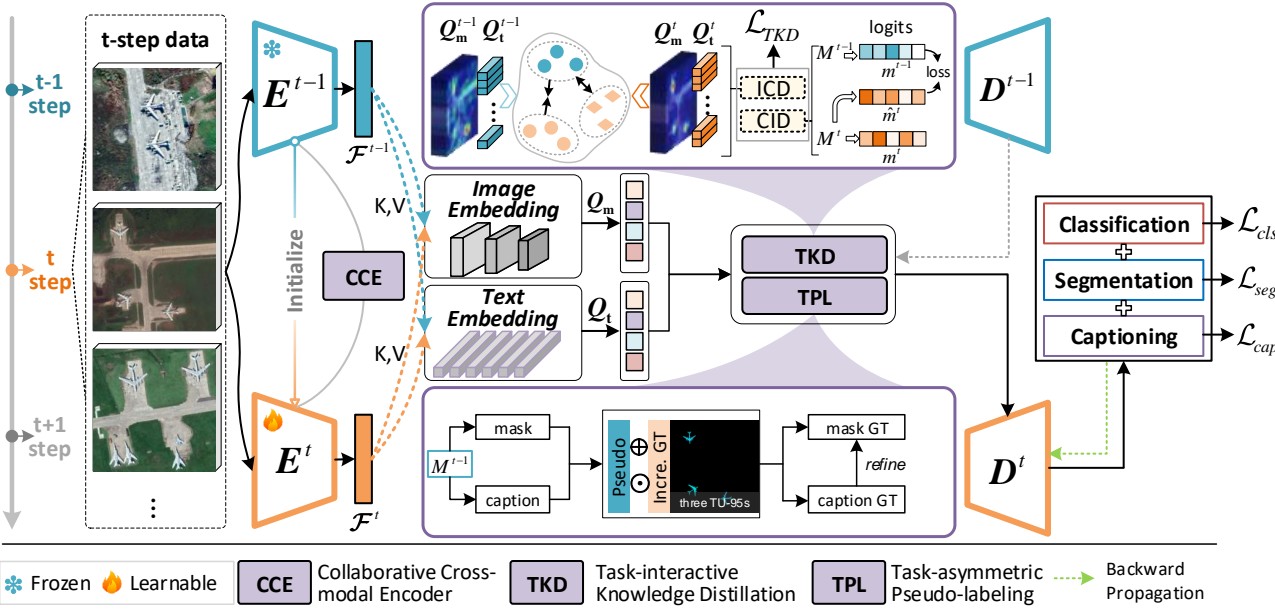

**Figure 2: Illustration of the proposed CPP network. The input consists of the incremental images with corresponding mask annotation and specific text-format annotation. The output consists of the mask predictions for both old and new classes and image captioning result with new semantics.**

Existing MTL studies mainly focus on feature-based and parameter-based methods. The former assumes that different tasks share a common feature representation. In deep learning, it learns a common feature representation for multiple tasks by sharing feature layers in a similar architecture [16, 48]. While parameters-based methods leverage parameter sharing [14, 41] across tasks, which mainly include hard sharing [5] and soft sharing [55]. The main challenges of MTL are the negative optimization and the *seesaw* phenomenon. Although MTL has been explored in incremental learning on classification [26], object detection [31] and image segmentation [49, 57], which still remains single-task interpretation. However, the multi-objective continual MTL still has a long way ahead. In this paper, we are committed to proving the feasibility of multi-task continual panoptic perception.

## 3 Method

As seen in Fig. 2, the proposed network consists of a Collaborative Cross-modal Encoder (CCE) for multi-modal feature extraction, a Task-interactive Knowledge Distillation (TKD) for model inheritance and a Task-asymmetric Pseudo-labeling (TPL) approach for retrieving old knowledge, respectively.

### 3.1 Preliminaries

Considering $\mathcal{D} = \{(x_i, y_i, r_i)\}$ as the training dataset, where $x_i \in \mathbb{R}^{C \times H \times W}$ and $y_i \in \mathbb{R}^{H \times W}$ denote the training image and mask annotation, respectively. $r_i$ is the captioning sentence. At $t$ step, $\mathcal{D}^t$ indicates the data for incremental training, $C^{0:t-1}$ indicates the previously learned classes and $C^t$ indicates the classes for incremental learning. When training on $\mathcal{D}^t$, the training data of old classes, i.e., $\{\mathcal{D}^0, \mathcal{D}^1, \cdots, \mathcal{D}^{t-1}\}$ is inaccessible. Using $M^{t-1}$ and

$M^t$ to represent the *t-1* and *t* step model, respectively. The total training process should consist of {Step-0, Step-1, $\cdots$, Step-T} steps.

### 3.2 Collaborative Cross-modal Encoder

In the proposed architecture, we aim to utilize a shared encoder for multi-task including dense prediction and image captioning. The former can be seen as a segmentation task and the latter is a global semantic understanding task. Thus we utilize a unified Transformer architecture to cope with cross-modal feature extraction.

As illustrated in Fig. 2, considering an image $x \in \mathbb{R}^{C \times H \times W}$, we firstly apply a model-agnostic feature extractor to extract image features. Concretely, the image encoder $E$ produces a downsampled feature $\mathcal{F} \in \mathbb{R}^{N \times H' \times W'}$. The image feature $\mathcal{F}$ derived from CCE can be utilized for both mask prediction and caption generation. Concretely, the encoded image feature from CCE is:

$$\mathcal{F} = \Theta(ReLU(\Theta(E(x)))) + E(x) \tag{1}$$

where $E(\cdot)$ is the encoding process. $\Theta(\cdot)$ is a linear mapping.

Whereafter the decoder is decoupled to a Transformer for mask prediction, one-to-one mappings between the mask predictions and the ground truth are generated. And a Transformer for word prediction. For pixel-level predication, the $\mathcal{F}$ is firstly generated to $N$ mask embeddings via a Transformer decoder with $N$ query input. We regard both instance and semantic segmentation as mask classification problems and handle them with a Transformer-based architecture. Concretely, assuming $N$ learnable queries $Q \in \mathbb{R}^{C_q \times N}$, $\mathcal{F}$ is used as keys (K) and value (V). A standard Transformer decoder is used to update $Q$. To make full use of shared features, the query is projected into mask embeddings $\mathbf{Q_m} \in \mathbb{R}^{N \times C_e}$ in the mask generation branch, and the text embedding $\mathbf{Q_t}$ for captioning.

Specifically for the captioning task, a standard Transformer decoder consisting of multiple decoder layers is used for sentence prediction, each having a masked self-attention mechanism, a cross-attention mechanism, and a feed-forward neural network. The self-attention mechanism can tackle the long-range context, which is beneficial to both segmentation and captioning tasks.

### 3.3 Task-interactive Knowledge Distillation

To inherit the capacity from the old model in the absence of old data, we propose a task-interactive knowledge distillation method under the circumstances of only the previous model $M^{t-1}$ and the incremental data $\mathcal{D}^t$ can be accessed. For fine-grained CL tasks, it faces misclassification challenges while the old classes and future classes are mixed in $bg$ class during CL steps. There are proofs showing the latent domain gap between the cross-modal data could lead to a learning ambiguity during knowledge distillation [42]. We propose a cross-task contrastive distillation method based on task-interactive guidance. Firstly, using $\mathbf{Q_m^{t-1}}$ and $\mathbf{Q_m}$ indicate the mask embeddings, while $\mathbf{Q_t^{t-1}}$ and $\mathbf{Q_t^t}$ for text embeddings. Concretely, the TKD module consists of Intermediate Contrastive Distillation (ICD) and Cross-guided Instance Distillation (CID).

**Intermediate Contrastive Distillation**. Since the segmentation task and the captioning task are both derived from the same features $\mathcal{F}$ produced by CCE, the ICD is conducted on both tasks. To alleviate the classifier confusion caused by semantic drift, we propose a contrastive distillation across $\mathbf{Q_m}$ and $\mathbf{Q_t}$. Inspired by [65], we argue that using embedding from the old model is more confident to reduce prediction error since catastrophic forgetting. Concretely, the optimization objective contains global output logits and class-wise contrastive learning between old and new classes. Thus the constraint for ICD is:

$$\mathcal{L}_{ICD} = d(\mathbf{Q_m^{t-1}}, \mathbf{Q_m^t}) + d(\mathbf{Q_t^{t-1}}, \mathbf{Q_t^t}) + D_{CL}(f_a^{t-1}, f_p^t, f_n^t) \quad (2)$$

$$D_{CL} = \frac{1}{|C^{0:t}|} \sum_{i,j,i\neq j}^{C^{0:t}} \sum_k^{C^{0:t-1}} \mathbb{1}[i=k][d(f_k^{t-1}, f_i^t) - d(f_k^{t-1}, f_j^t)] \quad (3)$$

where $d(\cdot)$ is a similarity measure to constraint the representation consistency between $M^{t-1}$ and $M^t$. $f_a^{t-1}$, $f_p^t$ and $f_n^t$ represent the *anchor*, *positive* and *negative* embeddings, respectively. $f_k^{t-1}$ indicates the embedding belonging to $k$-th class from $M^{t-1}$. $f_i^t$ and $f_j^t$ indicate the corresponding embeddings from $M^t$.

**Cross-guided Instance Distillation**. In the RSIs, the large amount of instances and complex distribution increase the probability of semantic fusion. Thus the instance segmentation requires fine-grained distillation. Directly distilling the feature map without considering foreground regions would result in training with a large amount of background information, leading to insufficient learning of important foreground regions and poor distillation performance. Thus we propose a distillation method emphasizing the interaction classification and localization, respectively. Since Chen et al. [12] propose a cross-modal instance distillation, we adapt it to multi-task CL scenarios. The quality score $q_r$ serves as an indicator to guide the student on which teacher's predictions should be paid more weight.

$$q_r = (c_r)^\gamma \times IoU(m_r^{t-1}, \hat{m}_r^t)^{(1-\gamma)} \quad (4)$$

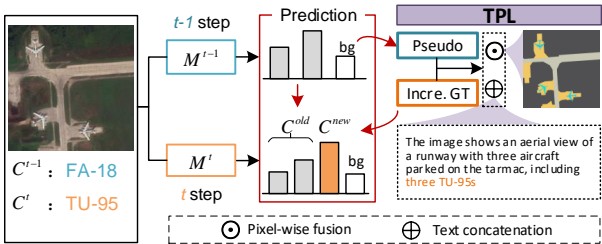

**Figure 3: Task-asymmetric pseudo-labeling. The asymmetric task reliance indicates the pseudo labels are cross-verified by more reliable predictions from multi-modal branches.**

where $\gamma$ indicates the weight of classification and segmentation. $c_r$ indicates the predictive classification results of instance $r$. $m^{t-1}$ and $\hat{m}^t$ are the segmentation results of $M^{t-1}$ and $M^t$, respectively. It is noted that $\hat{m}_r^t$ is the student output from the teacher decoder to restrain the model inheritance [51]. For fine-grained tasks, the recognition branch should be assigned a higher weight to alleviate misclassification due to large intra-class variance. Thus a constraint between $M^{t-1}$ and $M^t$ is proposed to optimize the instance segmentation task via a guided instance distillation loss.

The CID objective is defined as:

$$\mathcal{L}_{CID} = \sum_i [-q_r(d(y_r^{t-1}, y_r^t))] \quad (5)$$

where $y_r$ indicates the predicted classification result of instance $r$. Thus the TKD objective is the combination of ICD and CID:

$$\mathcal{L}_{TKD} = \mathcal{L}_{ICD} + \mathcal{L}_{CID} \quad (6)$$

### 3.4 Task-asymmetric Pseudo-labeling

Since only the incremental classes are labeled, we propose an exemplar-free pseudo-labeling method with asymmetric task reliance, i.e., more emphasis on confident predictions. As seen in Fig. 3, we observe the captioning task focuses more on the global context. While the segmentation tasks tend to cause pixel misclassification within a single intact target area. Using $\mathcal{Y}_{cap}^{t-1}$ indicates captioning output from the $M^{t-1}$. Thus the segmentation annotation $\bar{y}_i^t$ at $t$ step for pixel $i$ is defined as:

$$\bar{y}_i^t = \begin{cases} \tilde{y}_i^{t-1}, \text{if } (\tilde{y}_i^{t-1} \in C^{0:t-1}) \wedge [(p_i \geq \Gamma) \vee [(\tilde{y}_i^{t-1} \in \mathcal{Y}_{cap}^{t-1})] \\ y_i^t, \quad \text{if } (x_i \in C^t) \\ c^b, \quad otherwise \end{cases} \quad (7)$$

where $\tilde{y}_i^{t-1}$ is the pseudo label for pixel $i$ generated from $M^{t-1}$, $p_i$ is the predictive probability of pixel $i$ at $t$ step. $x_i$ is the $i$-th pixel in image $x$. $y_i^t$ is the incremental annotation for pixel $i$ at $t$ step. $c^b$ indicates the unknown background class. The asymmetry manifests itself by threshold or more rely on captioning predictions. Since the learned and future classes are mixed in $c^b$ at each CL step, the relative scoring of softmax could lead to significant semantic chaos and catastrophic forgetting during CL training steps. Instead, we take independent *Sigmoid* with binary cross-entropy loss to cope with the variable class number during CL steps and alleviate the interference from variable background semantics.

For the captioning task, every sentence starts with "START" and ends with "END" keywords. The label at $t$ step is concatenated with the pseudo label before "END". Thus the captioning label for $t$ step is:

$$\bar{y}_{cap}^t = \tilde{y}_{cap}^{t-1} \oplus y_{cap}^t \tag{8}$$

where $\tilde{y}_{cap}^{t-1}$ indicates the predicted captioning from $M^{t-1}$ and $y_{cap}^t$ is the readily available label at $t$ step. $\oplus$ indicates the concatenation operation.

## 3.5 Overall Objective via MTL

The proposed model supports end-to-end training with weighted losses across multi-objective joint training. Concretely, the objective considers four parts: classification loss $\mathcal{L}_{cls}$, segmentation loss $\mathcal{L}_{seg}$, caption generation loss $\mathcal{L}_{cap}$ and knowledge distillation loss $\mathcal{L}_{TKD}$. Considering joint training among diverse tasks, we propose a weighted loss for MTL issue. Concretely, $\mathcal{L}_{cls}$ is formed by a cross-entropy loss. And $\mathcal{L}_{seg}$ is binary mask loss:

$$\mathcal{L}_{seg} = \sum_{i=1}^{N} [\mathbb{1}[c_i^{gt} \neq \emptyset] \mathcal{L}_m(\hat{y}, \bar{y})] \tag{9}$$

where $\mathcal{L}_m$ is formed by focal loss [30] and dice loss [36]:

$$\mathcal{L}_m = \eta_1 \mathcal{L}_{focal}(\hat{y}, \bar{y}) + \eta_2 \mathcal{L}_{dice}(\hat{y}, \bar{y}) \tag{10}$$

where $\eta_1$ and $\eta_2$ are the weighting component. $c_i^{gt}$ is the label for class $c$. And the captioning loss is formed by a cross-entropy loss:

$$\mathcal{L}_{cap} = -\sum_{t=1}^{L} log p_{cap}(\bar{y}_{cap}) \tag{11}$$

where $p_{cap}$ is the word prediction probability from the caption module and $\bar{y}_{cap}$ represents the annotation. Thus the integrated objective is defined as:

$$\mathcal{L} = \mathcal{L}_{cls} + \mathcal{L}_{seg} + \lambda \mathcal{L}_{cap} + \mathcal{L}_{TKD} \tag{12}$$

where $\lambda$ is the weight of captioning tasks. Specifically, the distillation loss is only applied at CL steps.

## 4 Experiments

## 4.1 Datasets and Protocols

**Datasets**: FineGrip [64] is a multi-task remote-sensing dataset that includes 2649 images, with 12054 fine-grained instance segmentation masks belonging to 20 foreground classes, 7599 background semantic masks covering 5 classes, and 13245 fine-grained sentence description annotations. The sample number in the training set is much smaller than that in the validation set, which provides challenging but practical scenes for CL under limited data conditions. Note that all *stuff* classes are trained at the initial step since they are covered in most samples. We evaluate our model on 20-5 (2 steps), 15-5 (3 steps) and 15-2 (6 steps), respectively.

**Protocols**: There are mainly two different CL settings: *disjoint* and *overlapped*. In both settings, only the current classes $C^t$ are labeled and others are set as background. In the former, images at $t$ step only contain $C^{0:t-1} \cup C^t \cup C^{bg}$. While the latter contains $C^{0:t-1} \cup C^t \cup C^{t+1:T} \cup C^{bg}$, which is more realistic and challenging. In this study, we focus on *overlapped* setting. We also report two baselines for reference, i.e., *fine-tuning* on $C^t$, and training on all

classes *offline*. The former is the lower bound and the latter can be regarded as the upper bound of this task.

## 4.2 Implementation Details

We use MaskFormer [13] with ResNet-50 [18] as the base encoder to extract image features. For all experiments, the initial learning rate is 0.01 and decayed by a *poly* policy. The implementation is based on PyTorch 1.10 with CUDA 12.3 and all experiments are conducted on a workstation with four NVIDIA RTX 4090 GPUs. For all CL steps, the training epoch is set to 90. The hyper-parameters are set as $\lambda = 2.0$ according to the analysis in Sec. 4.4.1, $\eta_1 = 20.0$ and $\eta_2 = 1.0$. We compute the panoptic quality (PQ), segmentation quality (SQ) and recognition quality (RQ) to measure the segmentation efficiency. While the Bilingual Evaluation Understudy (BLEU) [38] is used to evaluate the generated text. The implementation is based on MMDetection [11].

## 4.3 Quantitative Evaluation

To comprehensively evaluate the CPP performance on old and new classes, we compute the segmentation performance and captioning performance at the initial step and the final step, respectively. The proposed model is tested on two aspects following [56]: multi-step with few-class (MSFC) and few-step with multi-class (FSMC). Particularly, FSMC emphasizes the ability to learn new knowledge (plasticity) since many new classes are adapted in a single step. In contrast, MSFC underlines the ability of anti-forgetting on old knowledge (stability) because many CL steps are conducted. The proposed model is evaluated from multi-modal CL performance and cross-task CL correlation, which are the key problems beyond the pioneer single-task CL approaches.

*4.3.1 Multi-modal CL performance.* As seen in Table 1, we first evaluate the PQ for old, new and all classes after all CL steps, respectively. Compared to naive fine-tuning approaches, freezing encoder after the initial step can help alleviate catastrophic forgetting. The proposed CPP method achieves superior anti-forgetting on old classes and plasticity on new classes simultaneously. The captioning performance is evaluated from two aspects: The performance at the initial step and after all CL steps. The former is the initial learned result with limited partial classes. While the latter indicates the matching accuracy of the predicted words after all CL steps. On the one hand, the captioning capacity improves since the new semantics are learned to meet with an intact understanding of the image. On the other hand, the BLEU scores declined after CL steps due to catastrophic forgetting.

The results in Table 1 also prove that the joint multi-task optimization in CL tasks should boost sub-task capacity. Compared to the segmentation method [13] in the proposed CL setting, the proposed CPP achieves 3.33%, 3.68% and 2.76% PQ improvements on all classes on 20-5, 15-5 and 15-2 tasks, respectively. While compared to fine-tuning method with freezing enoder, CPP achieves overall improvements across multi-modal tasks. Concretely, for 15-2 task, the long-step learning brings severe catastrophic forgetting. As a comparison, CPP maintains relative higher performance on both segmentation task and captioning task, which demonstrates the effectiveness of the proposed multi-modal continual learning architecture.

**Table 1: Quantitative performance on FineGrip dataset. We evaluate the segmentation task with PQ (%). $C^o$, $C^n$ and $C^a$ are the performance for old classes, new classes and all classes after all CL steps, respectively. The captioning performance is evaluated by BLEU scores (beamsize=5), which are reported before ($B^b$) and after ($B^a$) all CL steps. † indicates we re-implement the method to CL tasks. FE indicates *freezing encoder* manner.**

| Task | 20-5 (2 steps) | | | | | 15-5 (3 steps) | | | | | 15-2 (6 steps) | | | | |
|---|---|---|---|---|---|---|---|---|---|---|---|---|---|---|---|
| | $C^o$ | $C^n$ | $C^a$ | $B^b$ | $B^a$ | $C^o$ | $C^n$ | $C^a$ | $B^b$ | $B^a$ | $C^o$ | $C^n$ | $C^a$ | $B^b$ | $B^a$ |
| *fine-tuning* | 0.82 | 4.56 | 1.57 | 10.37 | 4.52 | 0.47 | 1.29 | 0.80 | 12.13 | 2.28 | 0.04 | 0.67 | 0.29 | 12.13 | 2.11 |
| *fine-tuning*-FE | 7.21 | 15.34 | 6.84 | 10.37 | 12.63 | 2.83 | 4.16 | 3.36 | 12.13 | 5.35 | 0.68 | 0.59 | 0.64 | 12.13 | 4.18 |
| MaskFormer† [13] | 23.29 | 30.98 | 24.83 | - | - | 25.73 | 22.19 | 24.31 | - | - | 14.25 | 7.82 | 11.68 | - | - |
| **CPP** | **27.06** | **32.59** | **28.16** | **35.93** | **34.12** | **29.71** | **25.41** | **27.99** | **33.01** | **27.00** | **17.20** | **10.30** | **14.44** | **33.01** | **21.52** |
| *offline* | 52.32 | 45.08 | 50.87 | 41.66 | 41.66 | 54.12 | 46.00 | 50.87 | 41.66 | 41.66 | 54.12 | 46.00 | 50.87 | 41.66 | 41.66 |

| Image | Segmentation GT | Pred. Before | Pred. After |
|---|---|---|---|
| The image features an aerial view of a military airfield, showcasing a row of nine FA-18 jets parked in a line. | | The aerial image shows an airfield with nine F-16s. | The aerial image shows an airfield with eight FA-18s. |
| Captioning GT | | Captioning Before | Captioning After |

| Image | Segmentation GT | Pred. Before | Pred. After |
|---|---|---|---|
| The aerial image shows a group of airplanes, with a total of 6 aircraft, consisting of four TU-160 bombers and two TU-95 bombers. | | The aerial image shows four aircrafts lined up on the runway, consisting of four TU-160s. | The aerial image shows six aircrafts lined up on the runway, consisting of four TU-160s and two Tu-95s. |
| Captioning GT | | Captioning Before | Captioning After |

**Figure 4: Qualitative visualization of the CPP *before* and *after* CL steps. The predictions are updated after CL steps on segmentation and captioning synchronously.**

**Table 2: Correlation study between multi-modal branches in 15-5 task.**

| Task | PQ | | | BLEU score | |
|---|---|---|---|---|---|
| | $C^o$ | $C^n$ | $C^a$ | before | after |
| Seg. only | 25.73 | 22.19 | 24.31 | - | - |
| Cap. only | - | - | - | 31.22 | 22.52 |
| CPP | **29.71** | **25.41** | **27.99** | **33.01** | **27.00** |

Fig. 4 depicts the CPP results on 15-5 and 10-5 tasks, respectively. It is noted from two aspects. On the one hand, the unknown semantics can be ignored at the previous steps, as seen in the first row in Fig. 4 before CL training. On the other hand, semantic chaos occurs when semantic instances are misclassified. As illustrated in the second row in Fig. 4, the foreground instances are located but misclassified to false class labels. While after the CL training steps by CPP, the semantic chaos is alleviated by distinguishing old and new classes. Meanwhile the captioning results are also replenished by taking into account both new and old semantics.

**Table 3: Ablation study of segmentation performance of the proposed method in PQ (%).**

| Method | 20-5 (2 steps) | | | 15-5 (3 steps) | | | 15-2 (6 steps) | | |
|---|---|---|---|---|---|---|---|---|---|
| | $C^o$ | $C^n$ | $C^a$ | $C^o$ | $C^n$ | $C^a$ | $C^o$ | $C^n$ | $C^a$ |
| *fine-tuning*-FE | 7.21 | 15.34 | 6.84 | 2.83 | 4.16 | 3.36 | 0.68 | 0.59 | 0.64 |
| +ICD | 24.32 | 31.08 | 25.67 | 27.86 | 22.53 | 25.73 | 15.01 | 9.54 | 12.82 |
| +CID | 22.19 | 26.53 | 23.06 | 24.61 | 19.42 | 22.53 | 14.89 | 6.92 | 11.68 |
| +ICD&CID | 26.79 | 32.06 | 27.84 | 28.15 | **25.44** | 27.07 | 16.33 | 9.17 | 13.47 |
| +TPL | **27.06** | **32.59** | **28.16** | **29.71** | 25.41 | **27.99** | **17.20** | **10.30** | **14.44** |

*4.3.2 Cross-task CL correlation.* In the proposed architecture, we utilize multi-modal CL branches to achieve CPP. We argue that mutual guidance from different branches in knowledge distillation and pseudo-labeling can improve each sub-task performance. To reveal the mutual impact between multi-modal branches, we perform separate training on the segmentation task and captioning task, respectively. As shown in Table 2, the joint training achieves 2.73% and 1.34% PQ improvement on the old and all classes after CL steps. It also achieves 4.48% BLEU score higher than the captioning branch alone after CL training steps. The results prove the mutual boosting of joint training across multi-modal tasks in CL problems.

## 4.4 Ablation Study

*4.4.1 Module Contribution.* To reveal the contribution of each module in the proposed method, we respectively disclose the corresponding modules as seen in Table 3. The proposed TKD combines segmentation flow and captioning flow to perform KD across different tasks. To disclose the effectiveness of TKD, the ICD and CID are separately and jointly verified, respectively. Compared to the fine-tuning method, ICD achieves significant improvement, proving the anti-forgetting effectiveness of leveraging intermediate features constraints between $M^{t-1}$ and $M^t$. While CID emphasizes the consistency of output distribution at CL steps to align the ability from the old model. On the other hand, the synergy of ICD and CID proves the homogeneity in the proposed CPP for multi-modal CL task. For example, the PQ on $C^a$ achieves 2.17% higher than ICD only and 4.78% higher than CID only on 20-5 task. While compared to the fixed-threshold-based pseudo-labeling method, the proposed TPL utilizes cross-task reliance to improve the confidence of the generated pseudo labels and achieves higher performance on all three CL scenarios. Note that when not adopting TPL, we apply a confidence-based pseudo-labeling with a fixed threshold with $\Gamma = 0.7$.

As defined in Eqn. 12, CPP utilizes a multi-task learning approach within a unified architecture. The weights assigned to multi-modal tasks significantly affect the efficiency of CL. As shown in Table 4, the performance of CPP achieves the highest PQ since the captioning task is weighted as $\lambda = 2.0$, demonstrating the cross-modal boosting in CPP tasks.

*4.4.2 Impact of base model.* The proposed architecture is model-agnostic, which supports various backbones and models. For the proposed CPP task, we propose a hypothesis that a strong base model can improve the CL efficiency. As seen in Fig. 5, various

**Table 4: Impact of hyper-parameters on 15-5 task.**

| $\lambda$ | 0.3 | 0.5 | 1.0 | 2.0 | 2.5 | 3.0 |
|---|---|---|---|---|---|---|
| PQ | 26.12 | 27.57 | 27.58 | **27.99** | 27.41 | 27.60 |

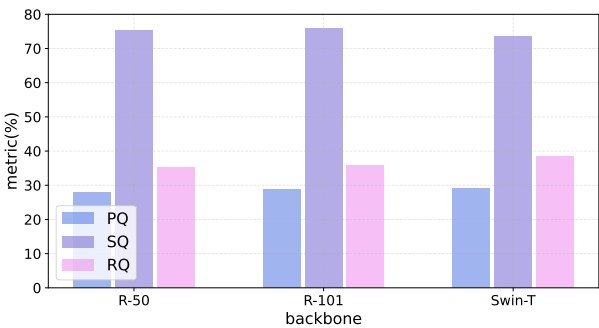

**Figure 5: Comparison of PQ, SQ and RQ on all learned classes after all CL steps with different backbones on 15-5 task.**

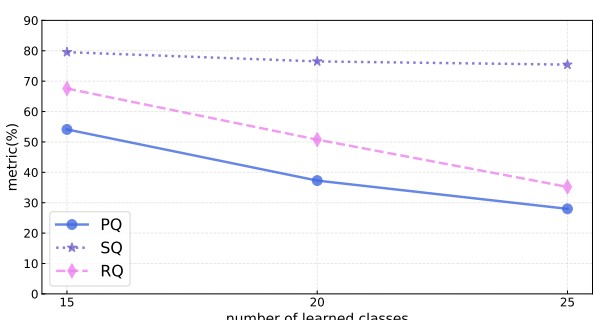

**Figure 6: The PQ, SQ and RQ evolution against the number of learned classes on 15-5 task.**

backbones including ResNet-50, ResNet-101 and Swin-T [32] are explored. Quantitatively, the model with ResNet-101 achieves higher PQ, SQ and RQ than that with ResNet-50, which proves a stronger backbone can achieve higher anti-forgetting performance and compatibility on new classes. It also shows that Transformer-based models achieve better recognition performance since the RQ score is significantly higher than two CNN-based models. However, CNN-based models achieve higher segmentation performance as the SQ

 Bo Yuan, Danpei Zhao, Zhuoran Liu, Wentao Li, and Tian Li

**Table 5: Comparison of various pseudo-labeling methods on 15-5 task.**

| Method | PQ | | |
|---|---|---|---|
| | $C^o$ | $C^n$ | $C^a$ |
| fixed ($\Gamma = 0.5$) | 26.46 | 24.65 | 26.34 |
| fixed ($\Gamma = 0.7$) | 28.15 | **25.44** | 27.07 |
| TPL | **29.71** | 25.41 | **27.99** |

scores are higher. We prefer it since the local features are more attentively utilized especially for small objects and areas.

*4.4.3 Compatibility on plasticity and stability.* With the incremental arriving data, the performance of old classes declined since lacking old annotations. To validate the anti-forgetting of the proposed CPP model, we evaluate the PQ, SQ and RQ evolution against the number of learning classes on 15-5 task. As seen in Fig. 6, the PQ and RQ performance of the old classes degenerates sharply due to catastrophic forgetting and semantic drift, which indicates the model degradation in pixel classification. While SQ maintains a high score since the background classes occupy the most pixels and being learned at the initial step. It indicates the anti-forgetting ability of recognition task in CPP task.

*4.4.4 Impact of pseudo-labeling.* Pseudo-labeling is an effective way to alleviate catastrophic forgetting due to the lack of old data and annotation. We compare the proposed TPL with the typical single-task confidence-based method with a *fixed* threshold, in which we set $\Gamma = 0.5$ as a convention and $\Gamma = 0.7$ following [17, 65]. As seen in Table 5, the proposed TPL achieves 1.56% PQ improvement on $C^o$ and 0.92% superiority on $C^a$ to the fixed confidence threshold setting ($\Gamma = 0.7$), which indicates the cross-task interactive can improve the pseudo-labeling effectiveness. However, the decline in $C^n$ shows the conflict between retaining the old knowledge and bias on the new knowledge.

*4.4.5 Robustness analysis.* To reveal the robustness to class learning orders of the proposed method, we perform experiments on 15-5 task with five different class orders including the ascending order and four random orders on *thing* classes as follows. In Fine-Grip [64] dataset, $C^{21-25}$ indicates the background stuff classes, which serves as the partial base classes in the CPP experiments.

$a : \{[21-25, 1, 2, 3, 4, 5, 6, 7, 8, 9, 10], [11, 12, 13, 14, 15], [16, 17, 18, 19, 20]\}$
$b : \{[21-25, 5, 7, 8, 9, 12, 14, 15, 16, 19, 20], [1, 2, 4, 11, 13], [3, 6, 10, 17, 18]\}$
$c : \{[21-25, 3, 4, 5, 8, 9, 13, 15, 17, 19, 20], [7, 10, 11, 16, 18], [1, 2, 6, 12, 14]\}$
$d : \{[21-25, 1, 2, 3, 11, 12, 14, 15, 16, 18, 20], [7, 8, 10, 17, 19], [4, 5, 6, 9, 13]\}$
$e : \{[21-25, 2, 3, 5, 7, 9, 10, 12, 13, 14, 19], [1, 4, 8, 16, 17], [6, 11, 15, 18, 20]\}$

The results shown in Table 6 indicate the PQ of $C^{1:10\&21-25}$ and $C^{11:20}$ on five different orders. Note that $C^{21-25}$ are background stuff classes that are learned at the initial step. The average PQ and standard variance are reported. It reveals that the class incremental orders have an evident impact on CPP performance. For example, the average PQ on $C^{1:10\&21:25}$ varies sharply. It proves the critical challenge of catastrophic forgetting in multi-task CL. On the other hand, the performance in all classes after all CL steps proves the learning stability of CPP.

**Table 6: Average performance on various class incremental orders on 15-5 task in terms of PQ (%).**

| order | $C^{1:10\&21-25}$ | $C^{11:20}$ | $C^{1:25}$ |
|---|---|---|---|
| a | 29.71 | 25.41 | 27.99 |
| b | 25.79 | 24.15 | 25.13 |
| c | 28.73 | 24.96 | 27.22 |
| d | 22.57 | 27.18 | 24.41 |
| e | 28.32 | 22.29 | 25.91 |
| avg.±std. | 27.02±2.58 | 24.80±1.60 | 26.13±1.32 |

**Table 7: Computational complexity in CPP task.**

| Task | Params. (M) | FLOPs (G) | FPS |
|---|---|---|---|
| Seg.-only | 45.0 | 126 | 8.5 |
| Cap.-only | 31.2 | 87 | 8.0 |
| CPP | 52.1 | 135 | 4.8 |

*4.4.6 Computational complexity.* To reveal the computational complexity of CPP, single-task CL approaches and multi-modal CPP are compared. The results in Table 7 come from 20-5 task with 800×800×3 input size after all CL steps. This indicates CPP implements multi-task and multi-modal CL with extra cost than single-task CL approaches.

*4.4.7 Limitations.* Considering CPP from multi-task learning perspective, the seesaw phenomenon can not be ignored since pixel-level segmentation and image-level captioning tasks have different training difficulty and convergence speeds. On the other hand, another potential improvement of CPP is to establish the coupling and mutual verification between the multi-modal information during CL steps.

## 5 Conclusion and Discussion

In this paper, we propose a continual panoptic perception (CPP) method that enables multi-task continual learning for multi-modal interpretation in remote-sensing images. Aiming to achieve CPP within a single intact architecture, the proposed architecture contains a shared encoder for multi-modal tasks, with cross-task distillation for solid knowledge inheritance. It is also proved that cross-modal task-interactive learning achieves mutual enhancement on sub-tasks. Experiments on the fine-grained panoptic perception dataset validate the effectiveness of the proposed method.

However, CPP encounters pendent challenges including intricate cross-modal feature relevance, unbalanced knowledge proportion and semantic chaos caused by catastrophic forgetting. Our future work will focus on improving joint optimization efficiency and developing more robust and interpretable CPP approaches.

## 6 Acknowledgments

This work was supported by the National Natural Science Foundation of China under Grant 62271018.

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
