# OpenReview forum: "Continual Panoptic Perception: Towards Multi-modal Incremental Interpretation of Remote Sensing Images"
_acmmm.org/ACMMM/2024/Conference — MM2024 Poster_

### Official Review · Reviewer_uDkk · 2024-05-07

**Rating:** 5
**Confidence:** 2

**Summary:**

The paper introduces the Continual Panoptic Perception (CPP) model, aiming to address challenges in continual learning. CPP comprises a Collaborative Cross-modal Encoder (CCE) for pixel-level classification and caption generation, Task-interactive Knowledge Distillation (TKD) to mitigate forgetting, and Task-asymmetric Pseudo-labeling (TPL) for knowledge retrieval. Experimental results validate the model's effectiveness across various protocols.

**Strengths:**

1. The proposed CPP model integrates a shared Encoder and modules for multi-task performance, including classification, segmentation, and captioning, showcasing innovative and logical design.
2. The efficacy of CCE and TKD is convincingly demonstrated in experiments, highlighting substantial performance gains with CPP.
3. This paper is well-written and easy to follow, which also shows its accessibility and readability.

**Limitations:**

1. I wonder which similarity measure the authors using in Equantion (3), is it cosine similarity or others? Clarification is needed on this similarity measure and its impact on results, as well as potential alternatives.
2. In the proposed Cross-guided Instance Distillation (CID) module, the authors utilize IoU to compute the quality score. Could you confirm whether this refers to the mean IoU (mIoU), which averages across different classes, or the class-agnostic IoU calculation?
3. Equation (4) references "c_i" without prior explanation, requiring inclusion of its definition for clarity.
4. The "PQ" used in the Abstract are encouraged to be replaced by the full name "Panoptic Quality" introduced in the main text, and here may cause some doubt without any explanation or the full name. Simultaneously, PQ, SQ and RQ are suggested to be illustrated and elaborated.

**Suitability:**

3

---

### Official Review · Reviewer_1pmN · 2024-05-21

**Rating:** 4
**Confidence:** 3

**Summary:**

This paper proposes a new multi-task learning architecture called CPP, which can solve the limitation of single task in typical CL. It can implement pixel-level classification, instance-level segmentation and image-level perception tasks in one model, and perform multimodal interpretation of remote sensing images. CPP also solves the common catastrophic forgetting and semantic drift problems in CL to a certain extent by using CCE, TKD and TPL, especially when old data is not available. The paper also conducts sufficient experiments on the proposed innovations.

**Strengths:**

1. Unlike the typical single-task CL method, the CPP framework proposed in this paper can implement multiple tasks in one model, covering pixel-level classification, instance-level segmentation and image-level perception at the same time.
2. The proposed TKD method effectively reduces the problem of catastrophic forgetting and improves the model's ability to retain previously learned knowledge by using cross-modal features for distillation.
3. The method proposed in the paper realizes end-to-end learning in a multi-task setting and simplifies the training process.
5. The experiments in the paper are very sufficient, and a relatively comprehensive ablation experiment is conducted on each parameter.

**Limitations:**

1. The data set selected in the paper is relatively simple, and the model's ability to be verified on multiple extensive data sets is not verified
2. The paper does not discuss the computational complexity of the method. The simultaneous processing of multiple tasks and multiple modalities may lead to high computational and memory requirements
3. The legend in Fig. 5 blocks part of the main figure, and even in conjunction with the content of Section 4.4.2 of the main text, this figure is not easy to understand.

**Suitability:**

2

---

### Official Review · Reviewer_mWZC · 2024-05-24

**Rating:** 3
**Confidence:** 3

**Summary:**

This paper proposed a unified continual learning model called CPP, that leverages multi-task joint learning covering pixel-level classification, instance-level segmentation and image-level perception for universal interpretation in remote sensing images. Concretely, they proposed a collaborative cross-modal encoder (CCE) to extract the input image features, which supports pixel classification and caption generation synchronously. To inherit the knowledge from the old model without exemplar memory, they proposed a task-interactive knowledge distillation (TKD) method, which leverages cross-modal optimization and task-asymmetric pseudo-labeling (TPL) to alleviate catastrophic forgetting. Furthermore, they also proposed a joint optimization mechanism to achieve end-to-end multi-modal panoptic perception.

**Strengths:**

The Figures in the manuscript are exquisite.

**Limitations:**

1.	The manuscript could benefit from a more comprehensive comparative analysis with state-of-the-art approaches. Specifically, it lacks a detailed discussion on how the proposed CPP model performs relative to existing models in handling catastrophic forgetting and semantic drift, critical challenges in continual learning.
2.	In lines 739-740 of the manuscript, the authors should elaborate further on why the threshold is fixed at 0.7 when TPL is not used.
3.     In Table 1, the segmentation task with SQ and RQ also needs to be evaluated and added.

**Suitability:**

2

---

### Meta-Review · Area_Chair_HUvk · 2024-06-30

**Recommendation:** Accept (Poster)
**Confidence:** 5

**Metareview:**

This paper proposes the Continual Panoptic Perception (CPP) model, a novel continual learning framework aimed at enhancing multi-modal incremental interpretation of remote sensing images.

After rebuttal and discussions, this paper receives two positive ratings and one negative rating. After carefully  reviewing the paper and all the reviewers' comments, The AC agrees to accept the paper and strongly recommends incorporating the content from the rebuttal into the final version.